# Seawater Hydration Modulates IL-6 and Apelin Production during Triathlon Events: A Crossover Randomized Study

**DOI:** 10.3390/ijerph19159581

**Published:** 2022-08-04

**Authors:** Olivia González Acevedo, Jerónimo Aragón-Vela, Juan Carlos De la Cruz Márquez, Manuel Martínez Marín, Rafael A. Casuso, Jesús R. Huertas

**Affiliations:** 1Institute of Nutrition and Food Technology “José Mataix”, Biomedical Research Center, University of Granada, 18106 Granada, Spain; 2Department of Physiology, School of Pharmacy, University of Granada, Campus de Cartuja s/n, 18071 Granada, Spain; 3Department of Physical Education and Sport, Faculty of Sport Sciences, University of Granada, 18001 Granada, Spain; 4Department of Health Sciences, Loyola Andalucía University, 41704 Sevilla, Spain

**Keywords:** exercise physiology, endurance exercise, sweating, hydration

## Abstract

A triathlon is an endurance event in which athletes need an efficient hydration strategy since hydration is restricted at different stages. However, it seems that seawater intake can be a suitable hydration alternative for this type of endurance event. Therefore, the aim of this study was to evaluate the efficacy of seawater hydration during a triathlon on cytokine production. Fifteen trained male triathletes (age = 38.8 ± 5.62 years old; BMI = 22.58 ± 2.51 kg/m^2^) randomly performed three triathlons, one of them consuming seawater (Totum SPORT, Laboratories Quinton International, S.L., Valencia, Spain), the other one consuming tap water ad libitum, and the last a physiologic saline solution as placebo. The triathlon consisted of an 800 m swim, a 90 km bike ride, and a 10 km run. Blood samples were taken at rest and after training, where markers of inflammation, hemoglobin, and hematocrit concentration were assessed. While the seawater was not ergogenic, it significantly increased the release of IL-6 and apelin post-exercise. However, no differences were found between the fractalkine, IL-15, EPO, osteonectin, myostatin, oncostatin, irisin, FSTL1, osteocrin, BDNF, and FGF-21 values over those of the placebo group. The present study demonstrates that hydration with seawater stimulates myokine production, which could lead to improved performance recovery after exercise.

## 1. Introduction

A triathlon is an endurance event where athletes have limited access to fluid intake. Particularly in the swimming and running stages, due to their dynamic nature, fluid intake is restricted. For this reason, an efficient hydration strategy must be established to avoid a significant negative impact on both aerobic performance [1] and athlete health [2] due to progressive dehydration combined with hyperthermia [3]. In addition, the amount of fluid consumed should be carefully considered. Indeed, excessive water consumption may induce a state of hyponatremia [4] and rehydration with an electrolyte-enriched drink may reduce susceptibility to muscle cramps [5]. Therefore, both the volume of the rehydration fluid and its composition are critical for maintaining whole-body fluid homeostasis. Currently, a hydration alternative that is gaining popularity in the sports community is the intake of seawater (SW) [6], given that deep ocean mineral water contains components which could complement and increase human recovery following an exhaustive physical challenge [7].

Although water is sufficient to rehydrate after non-strenuous exercise training, for athletes it may not be adequate due to the increased loss of electrolytes. Indeed, large sweat losses induce significant salt elimination; for example, an athlete who sweats about 5 L will eliminate between 4600 and 5750 milligrams (mg) of sodium. Therefore, an adequate rehydration programme must restore electrolyte losses [8]. Stasiule et al. reported that ingestion of deep mineral water could accelerate recovery of aerobic capacity and leg muscle power compared with ingestion of tap water alone [9]. In addition, it has been shown that desalinated ocean mineral water, taken from 662 m below sea level, can substantially accelerate recovery of aerobic power and lower-body muscle power after a prolonged bout of dehydrating exercise [7]. SW is mainly characterized by a high percentage of minerals such as magnesium (Mg^2+^). Indeed, Mg^2+^ deficiency may potentiate exercise-induced muscle damage and stress response, as well as exacerbate the inflammatory state due to an increase in substance P-mediated secretion of pro-inflammatory cytokines [10]. However, the evidence for the beneficial effects of SW is still limited, as it has not been thoroughly evaluated in endurance sports, such as triathlon, with the compounding factor that participants have limited access to liquids.

Several studies suggest that post-competition triathletes exhibit a marked state of mild to moderate hypohydration [11,12]. This might be because not enough fluid has been ingested to compensate for sweat losses during exercise [13] which, combined with hyperthermia, could induce cardiovascular instability, such as reduction in plasma volume, cardiac filling, and stroke volume [3,14]. Therefore, this state of dehydration could lead to a stress response and, thus, probably a suppression of the immune function [15], even increasing exercise-induced inflammatory responses [16]. Particularly, the dehydration status could increase the systemic response of cytokine production and a state of transient imbalance between the pro-inflammatory and anti-inflammatory cytokines in the system [15], which could negatively interfere with the regeneration of damaged tissue after exercise [14]. These so-called cytokines are produced in response to exercise and their release may improve post-exercise recovery [17,18]. One of the most influential post-exercise cytokines may be IL-6; indeed, after an endurance event, circulating IL-6 can rise to 100 times [19]. It is accepted that the elevated IL-6 resulting from endurance exercise is released directly from skeletal muscle and promotes liver glycogen breakdown and lipolysis of fatty acids in adipose tissue [20]. In fact, there appears to be a link between SW consumption and IL-6 release, as Lu et al. reported that SW consumption affects IL-6 expression in the heart of diabetic rats [21]. Therefore, this suggests that SW can alter the circulating levels of IL-6 in response to a triathlon.

In summary, the authors of this study hypothesize that SW consumption could be a suitable hydration alternative, which could improve post-exercise recovery periods, mainly in endurance events. Therefore, the main purpose of the present study was to evaluate the efficacy of a hydration strategy through SW consumption in a triathlon event under natural environmental conditions. In addition, as a secondary aim, we wanted to test whether SW consumption could alter myokine release after exercise.

## 2. Materials and Methods

### 2.1. Subjects and Ethics

The protocol was conducted according to the participation criteria, which included having an annual competitive calendar, having completed at least 3 triathlons one month before the study, obtaining written informed consent, and the complete absence of metabolic diseases, acute cardiovascular diseases, and current infectious diseases. The present experimental crossover study involved observing fifteen federated triathletes from different clubs in Andalusia (Spain). The participants in this study had completed an average of 11.3 ± 5.77 years of training, with a weekly training session schedule of 4.25 ± 1.37 days/week, with an average of 66.25 ± 19.93 km/bike. Exclusion criteria were inflammatory diseases in the last three months, cigarette and alcohol consumption, diabetes, and cardiorespiratory pathologies such as asthma and allergies. After receiving detailed information on the objectives and procedure of the study, each participant signed an informed consent form for participation, which complied with the ethical standards of the World Medical Association Declaration of Helsinki (2013). The study was approved by the local Ethics Committee the University of Granada (209/CEIH/2018).

### 2.2. Experimental Design

A double-blind, placebo-controlled, and crossover experimental study with pre- and post-test measurements was carried out. We randomly divided the 15 participants into three groups. The experimental group (SG) consumed seawater and the placebo group (PG) consumed a physiologic saline solution (9 g/L NaCl), whereas as for the control group (CG), a third trial was conducted with only 10 participants who consumed tap water ad libitum. Each experimental group performed a complete triathlon test with its three modalities. Participants visited our facilities on four occasions to complete the experimental design. For the morning session, participants came to the swimming pool after a standardized breakfast (57% carbohydrate, 18% protein, and 25% fat; thus, the breakfast met the general recommendations of international institutions) eaten at least one hour before their arrival, and then the subjects rested passively for 10–15 min before we recorded their basal measurements. The order of the hydration strategy was randomized. The ampoules for SG contained a commercial saline product with buffered electrolyte salts (Totum Sport, Laboratories Quinton International, S.L., Valencia, Spain). The total amount of electrolytes supplied to SG were 27.297 mg/L sodium, 0.465 mg/L Potassium (K^+^), and 19.5 mg/L Mg^2+^, 1.377 mg/L Calcium (Ca^2+^). The SW treatment was carried out according to Pérez–Turpin et al. [22]. The amount of salt was calculated to replace ~50% of the sodium loss by sweating for an average participant. The PG received the same number of drinkable ampoules with exactly the same appearance but filled with a physiologic saline solution, in order to have the same salty taste as the experimental sample. No strenuous exercise was allowed 72 h before each protocol and subjects followed the same dietary recommendations. This study was conducted on the facilities of the Faculty of Sports Science of the University of Granada.

### 2.3. Experimental Trial

Before the test, with the collaboration of a nutritionist, the participants were instructed to ingest food and drink, 72 h in advance, at least 8.5 g of carbohydrate per kilogram of body weight and at least 50 mL/kg^−1^ of fluid [23]. We also asked them to avoid any source of caffeine and alcohol. The purpose of the strictly controlled diet was to obtain elevated muscle and liver glycogen stores. At the first visit, the VO_2max_ of the participants was determined. For this purpose, we used a graded exercise test to exhaustion on an electronic brake (Lode Excalibur sports ergometer, Groningen, The Netherlands) with breath-by-breath measured fractions of inspired and exhaled O_2_ and CO_2_. The graded exercise protocol was initiated at 150 W and increased by 30 W × 2 min^−1^ until voluntary exhaustion or a pedaling frequency of >75 rpm [24]. On the second visit, participants carried out the first test. Before starting the test, participants drank 500 mL of tap water and performed 10 min of a standard warm-up routine consisting of running, dynamic leg exercises, and jumping practice. Participants received three plastic bags with four translucent drinkable ampoules each. All participants were instructed to ingest the contents of the first ampoule during the transition between swimming and cycling, the second ampoule approximately halfway through the cycling, and the third ampoule during the transition between cycling and running. The empty capsules had been returned to the researcher, in order to check that the capsules were consumed. This protocol was adopted to assure the proper distribution of hydration times during the triathlon to allow electrolyte intake and absorption during the event. The test consisted of an 800 m swimming exercise, 90 km cycling (net gain of 1100 m altitude), and 10 km running. On the third visit, within 7–14 days of the previous test, the same one was performed, but with a different hydration strategy. Finally, the CG conducted a third trial, where only tap water was consumed ad libitum. Immediately after the running test, a blood sample and anthropometric data was taken in all groups. In addition, immediately after each transition, fingertip blood samples were used (5 μL) to obtain hematocrit, hemoglobin, and lactate values.

### 2.4. Anthropometric and Physiological Variables

Fifteen minutes prior to the competition and within one minute after the end of the running test, participants were weighed in running clothes with a segmental bioelectrical impedance analyzer (InBody 720) (Biospace Inc., Seoul, Korea). Heart rate (HR) (PE-3000 Sport-Tester, Polar Inc., Kempele, Finland) and systolic and diastolic blood pressure (SBP/DBP) (HEM-907 XL, Omron Corporation, Kyoto, Japan) were also measured. Heart rate was monitored with a Polar Heart Watch system (Polar Electro Inc., Lake Success, New York, NY, USA) without interruption during the test. The HR_max_, HR_mean_, and Hr_post_ during the different stages of the test were calculated with the software provided by Polar.

### 2.5. Sample Collection

Fifteen minutes before the competition and within one minute after the end of the race, participants were placed in reclining chairs while 2 mL of venous blood was taken by venipuncture from the antecubital vein in vacutainers without an anticoagulant. Blood samples were appropriately coded, and the serum was isolated by centrifugation 3500 rpm (1137× *g*) and stored at −80 °C in a controlled deep freezer until use for bulk biomarker analysis. Serum separation was performed within 2 h of collection using appropriate facilities and standard operating procedures for handling biological samples.

### 2.6. Hematological Measurements

At baseline and at the end of each exercise, blood lactate was measured (Lactate Pro, Kyoto, Japan) using fingertip blood samples (5 μL). Hematocrit was obtained by microcentrifugation for 10 min at 11,000 rpm (BIOCEN). Hemoglobin was analyzed by Drabkin’s method, whereby 20 μL samples of blood were oxidized and quantified spectrophotometrically at 540 nm. Changes in plasma volume at post were calculated using the Dill and Costill [25] equation as follows:ΔPV (%) = 100 × ((Hbpre/Hbpost) × (100 − HCTpost)/(100 − HCTpre) − 1)

### 2.7. Cytokine and Myokine Proteins Multiplex Analysis

Biomarkers were analyzed by multiplex techniques using the Magpix kit (Millipore, Burlington, MA, USA) according to the SP manufacture’s recommendations. For these techniques, the commercial Human Myokine Magnetic Bead Panel kit (HMYOMAG-56K) was used, which is validated for use in human samples and has a calibration line that allows for inferring the loading of molecules in serum. The biomarkers to be determined were the following myokines and cytokines: apelin, fractalkin, brain-derived neurotrophic factor (BDNF), erythropoietin (EPO), osteonectin, leukemia inhibitory factor (LIF), IL-15, myostatin (MSTN), fatty-acid-binding protein 3 (FABP-3), irisin, fibroblast growth factor 21 (FGF21), folstn-like1Pro (FSTL-1), oncostatin M, IL-6, and osteocrine/musclin.

### 2.8. Statistical Analysis

Data are expressed as mean ± standard error of the mean. Basal level of myokines is presented as the mean of the three pre-tests. The normality of distribution was assessed by the Shapiro–Wilk test. The homogeneity of variance was analyzed through the Levene test. A two-way ANOVA with repeated measures was used to reveal the effects of SG vs. PG and CG and pre-exercise vs. post-exercise differences between the groups. Multiple comparisons were performed by the Bonferroni post-hoc test. The level of significance was set at *p* < 0.05. All statistical procedures were carried out using SPSS/PC V. 22 (SPSS Inc., Chicago, IL, USA).

## 3. Results

### 3.1. Participants’ Anthropometric and Physiological Characteristics

Table 1 shows the age, anthropometric characteristics, and peak oxygen consumption data of the participants. No baseline differences were found between the groups that were compared. The blood values, blood pressure, heart rate, effort, and performance of each group in the different stages of the test were shown in Table 2. No significant differences were found when the experimental groups were compared during transitions. However, after the running and the cycling stages, the CG had increased hematocrit if compared with their baseline values.

### 3.2. Alterations in Blood Cytokine and Myokine Protein Levels

In order to detect any SP dysregulation of post-exercise molecule release after SW intake, several cytokines and myokines were analyzed. SW or placebo consumption was well accepted, and participants reported no side effects. Plasma levels of apelin had significantly increased in SG compared with BG (*p* < 0.001), while no significant changes in plasma levels of apelin in CG and PG compared with BG (Figure 1D) were observed. Furthermore, when comparing means between groups, the increase in apelin in SG vs. GC and PG was also significant (*p* < 0.001) (Figure 1D).

Similarly, plasma levels of BDNF were increased in SG compared to BG (*p* = 0.015), but BDNF levels were not increased in CG nor in PG (Figure 1A). Furthermore, the plasma levels FABP3 showed a significant increase in CG (*p* < 0.001), SG (*p* < 0.001), and PG (*p* < 0.001) compared to BG (Figure 1C). Post-test measurement of plasma levels of FGF21 showed an increase in all groups compared to BG; however, significant differences were only obtained with the SG (*p* = 0.016) (Figure 2C).

In Figure 2, in the case of IL-6, post-training plasma levels were significantly increased in the SG (*p* < 0.001) compared to the BG, while PG and CG did not show any post-test increase. (Figure 2A). Furthermore, when the means were compared between groups, the difference between the expression of SG vs. GC and SG vs. PG (*p* < 0.001) were significant in both cases (Figure 2A).IL-15 was significantly increased after exercise in the groups receiving both PG (*p* < 0.001) and hydration with SG (*p* = 0.011). However, the CG maintained baseline values (Figure 2C). As for the LIF response, a significant post-exertion decrease was shown exclusively in the CG (*p* = 0.042) (Figure 2B). In the triathlon session there were no differences between pre- and post-exercise measurements, nor between groups, in the plasma levels of the molecules fractalkine, irisin, EPO, osteonectin, MSTN, (FSTL-1), oncostatin M, and osteocrine/musclin (Table 3).

## 4. Discussion

The main purpose of the present study was to evaluate the effect of a seawater hydration on body composition, performance, and circulating myokine proteins during a triathlon. While no effects were observed in performance and body composition outcomes, we found that the release of several myokines is strongly modulated by SW.

### 4.1. Anthropometric, Performance, and Effort Parameters

All of the experimental groups performed an adequate hydration pattern as denoted by unchanged post-exercise weight [26]. This is in line with a study showing similar post-exercise weight between SW and placebo following 60 min of running at 75% of VO_2_max [27]. Therefore, it makes sense that when compared with other hydration procedures the results presented here and those reported by others [6,7,9] show no ergogenic effects of SW. Additionally, we show that SW hydration does not alter lactate concentration following a triathlon. This observation is in contrast with a study showing a lower lactate concentration after a running exercise in the SW condition compared to pure water [22]. However, this study failed to properly report the intensity of each exercise, making it difficult to ensure that both hydration conditions were performed at the same relative intensity. In this regard, we observed similar HR_mean_ and HR_post_ during the different hydration conditions, indicating that at similar relative intensity neither performance nor circulating lactate is affected by SW consumption.

### 4.2. Cytokines, Myokines, and Hormones Values

One of the most important observations of the study is that SW stimulates an early release of IL-6. IL-6 has a canonical inflammatory effect when secreted by macrophages, such as during chronic sedentarism [28]. However, if secreted during exercise, it can have an anti-inflammatory effect [20]. In addition, exercise-induced IL-6 could increase glucose uptake and mitochondrial content within skeletal muscle through modulation of AMPK activity [29,30,31]. Notably, in vitro studies have suggested that SW can stimulate mitochondrial biogenesis through AMPK activation [32,33]. Therefore, our data might suggest that SW could stimulate skeletal muscle’s oxidative metabolism by a longer exposure of the muscle cells to available IL-6. Although this hypothesis needs to be further tested, it is also supported by the fact that apelin follow a similar release response after SW hydration. Indeed, both molecules are strongly related with oxidative metabolism through AMPK [34,35,36]. Therefore, an extended period of IL-6 release, would induce an increase in the repair window and facilitate AMPK activity via an AMP-independent mechanism. However, although IL-15, BDNF, and FGF21 showed significant differences when comparing pre- vs. post-exercise groups, no differences were found when comparisons were made between all other groups. Therefore, we believe that this significant production enhancement was not due to SW consumption but rather to the sporting event itself.

In contrast, we found that other myokines and hormones have not been significantly changed either by the exercise protocol or by SW consumption. For instance, fractalkine, EPO, osteonectin, myostatin, oncostatin, irisin, FSTL1, and osteocrin, did not show any type of modulation after exercise when baseline values were compared with the three groups analyzed [37,38,39]. It is possible that these parameters were not affected because most of these molecules are involved in the development of the central nervous system, cellular homeostasis, and bone, and therefore endurance exercise would not be a sufficient stimulus to initiate any changes. In the case of irisin, it has been questioned whether it can even be an exercise-induced protein as it has not been undoubtedly proven if exercise can induce its transcription [40]. The case of LIF is a little bit different, as it has similar roles to IL-6 [41]; thus, the fact that it is only induced in the CG group could reflect that the protocol itself is not associated with IL-6 release, while the net significant (SW) and numerical (PG) increase of IL-6 would prevent LIF release.

### 4.3. Limitations and Strength

Several limitations of this study must be highlighted. (i) The main limitation of this study is that we did not have access to muscle samples to analyze muscle AMPK activity and/or mitochondrial function markers. (ii) The triathlon modality used is not within the official modalities, given that our aims were related more to the hydration capacity of SW consumption, rather than the triathlon modality itself. (iii) The third limitation is the number of triathletes recruited. It would have been interesting to have a greater number of triathletes of both sexes in order to have a greater statistical power and to obtain results with gender distinction. However, the main strength of this study is that it opens a new field of research where seawater can be used to manage exercise-induced cytokine release. In fact, further studies are needed in order to investigate whether SW can potentially improve skeletal muscle’s oxidative metabolism in athletes and diseased populations.

## 5. Conclusions

Acute SW consumption as a hydration strategy does not have a significant ergogenic effect in triathletes. However, the fact that IL-6 and Apelin, closely related to oxidative metabolism, are potentially released by SW opens a new field of research. In addition, it is necessary to test whether the chronic intake of SW can be ergogenic by improving the oxidative capacity of skeletal muscle.

## Figures and Tables

**Figure 1 ijerph-19-09581-f001:**
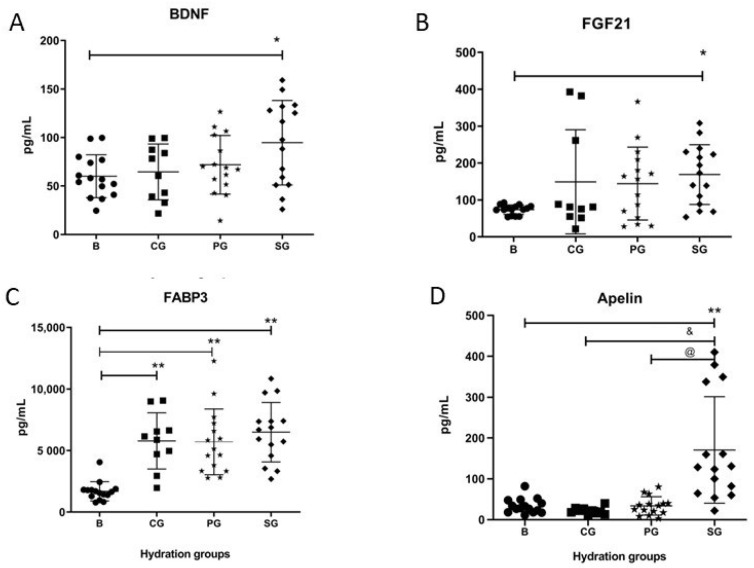
Expression of myokine proteins related to glucose metabolism and synaptic function. (**A**) The BDNF increased at increased post-exercise in experimental group (SG). (**B**) FGF 21 expression values showed a significant increase post-exercise only in the SG. (**C**) FABP3 expression values change in each groups post-exercise. (**D**) The apelin increased at increased post-exercise in SG. CG, Control Group; BG, Basal; PG, Placebo group. Each participant is represented with a symbol. Data are shown as means ± SD. * *p* < 0.05 basal vs. post-exercise of each group. ** *p* < 0.001 basal vs. post-exercise of each group. @ *p* < 0.05 SG vs. PG. & *p* < 0.05 SG vs. CG.

**Figure 2 ijerph-19-09581-f002:**
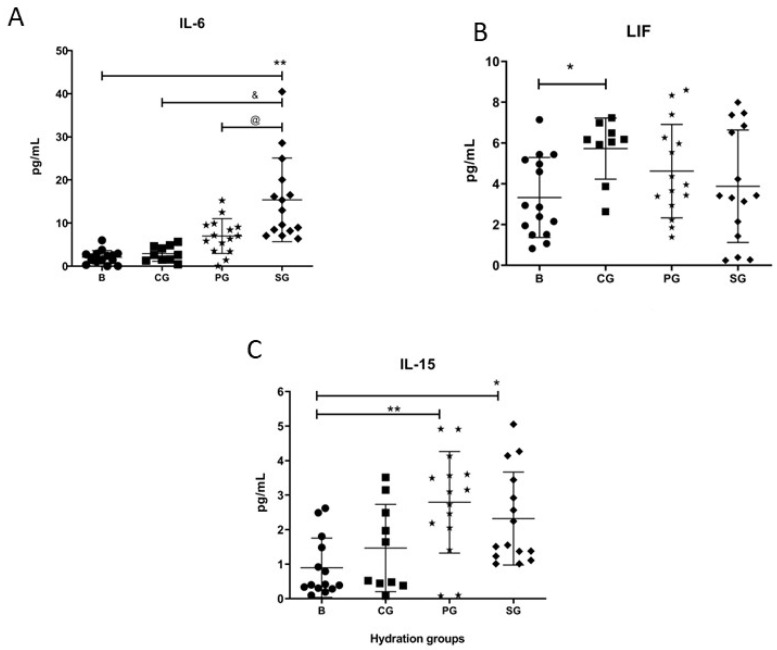
Expression of myokine proteins. (**A**) The IL-6 levels just increased post-exercise in experimental group (SG). (**B**) Leukemia Inhibitory Factor (LIF) values increased post-exercise only in the SG. (**C**) IL-15 values increased in both the placebo group and the post-exercise SG. CG, Control Group; B, Basal; PG, Placebo group. Each participant is represented with a symbol. Data are shown as means ± SD. * *p* < 0.05 basal vs. post-exercise of each group. ** *p* < 0.001 basal vs. post-exercise of each group. @ *p* < 0.05 SG vs. PG. & *p* < 0.05 SG vs. CG.

**Table 1 ijerph-19-09581-t001:** Participants’ characteristics (mean, SD).

	Control (*n* = 10)	Experimental—Placebo (*n* = 15)
**Age (years)**	40.70 ± 5.89	38.80 ± 5.62
**Body height (cm)**	176.60 ± 8.04	176.00 ± 6.43
**Body mass (kg)**	70.74 ± 12.28	73.26 ± 10.68
**Total Body Water (kg)**	46.54 ± 6.58	46.17 ± 5.16
**Protein (kg)**	12.72 ± 1.84	12.55 ± 1.39
**Body Fat Mass (kg)**	7.19 ± 3.92	10.22 ± 5.29
**Body Mass Index (kg/m^2^)**	22.58 ± 2.51	23.54 ± 2.34
**VO_2_max (mL/kg/min)**	45.31 ± 9.52	43.46 ± 10.43

**Table 2 ijerph-19-09581-t002:** Heart rate response, hematological measurements, and stress data.

				Swim									Bike									Run								
Variables	Basal			PG			CG			SG			PG			CG			SG			PG			CG			SG		
**Haematocrit**	46.6	±	3.41	45.80	±	3.55	47.50	±	3.06	45.93	±	3.71	46.40	±	3.48	47.5 *	±	2.95	45.47	±	3.07	45.20	±	3.03	47.88 ^#^	±	2.09	46.33	±	3.11
**Haemoglobin**	12.2	±	3.10	11.73	±	3.45	14.23	±	1.36	11.79	±	1.74	12.28	±	3.75	14.69	±	1.42	11.09	±	1.94	11.55	±	1.28	15.16	±	2.71	12.04	±	1.81
**Hrmax (bpm)**	xxx	±	xxx	143.00	±	7.74	165.00	±	8.19	145.40	±	6.02	181.67	±	5.29	160.80	±	11.7	176.93	±	4.70	173.93	±	6.24	170.40	±	9.61	177.87	±	8.45
**Hrmean (bpm)**	61.9	±	10.5	130.49	±	34.7	139.10	±	10.3	135.06	±	7.13	162.98	±	42.9	137.70	±	10.5	135.06	±	7.13	162.98	±	42.91	155.10	±	9.93	149.30	±	7.19
**Hrpost (bpm)**	xxx	±	xxx	105.93	±	14	99.40	±	19.2	108.73	±	15.5	181.67	±	5.29	149.90	±	7.26	150.33	±	8.64	169.67	±	5.74	171.10	±	3.21	175.27	±	9.16
**SBP (mmHg)**	127	±	12.6	xxx	±	xxx	xxx	±	xxx	xxx	±	xxx	xxx	±	xxx	xxx	±	xxx	xxx	±	xxx	117	±	15.30	110 *	±	11.3	122	±	15.4
**DBP (mmHg)**	74	±	8.44	xxx	±	xxx	xxx	±	xxx	xxx	±	xxx	xxx	±	xxx	xxx	±	xxx	xxx	±	xxx	68	±	7.93	66	±	4.45	70	±	11.5
**Lactate (mmol/L)**	1.68	±	0.36	5.85	±	1.68	7.10	±	2.66	5.86	±	2.22	2.47	±	1.57	2.46	±	1.16	3.19	±	1.85	2.83	±	1.59	2.92	±	0.95	2.71	±	0.62
**RPE**	xxx	±	xxx	7.05	±	0.68	7.94	±	0.93	6.70	±	0.67	7.80	±	0.42	8.40	±	0.54	8.00	±	1.30	9.00	±	0.81	8.87	±	0.54	9.15	±	0.88
**Perfor. (min)**	xxx	±	xxx	14.68	±	1.45	14.55	±	1.39	14.55	±	1.24	165.70	±	12.9	157.20	±	8.02	165.60	±	12.8	51.13	±	9.59	48.61	±	6.46	51.95	±	7.95
**Weight (kg)**	71.8	±	12.0	xxx	±	xxx	xxx	±	xxx	xxx	±	xxx	xxx	±	xxx	xxx	±	xxx	xxx	±	xxx	70.86	±	11.63	69.15	±	11.6	70.98	±	11.5

Results are expressed as mean ± standard deviation (SD). SBP, Systolic blood pressure; DBP, Diastolic blood pressure; PG, Placebo Group; CG, Control Group; SG, Experimental Group; BP, Blood pressure. *: *p* > 0.05 vs. rest; ^#^: *p* > 0.001 Experimental Group (SG) vs. Control Group (CG).

**Table 3 ijerph-19-09581-t003:** Myokines data.

*Myokines*	Rest			Control Group			Placebo Group			Experimental Group		
**Fractalkine**	419.70	±	194.30	435.3	±	239.00	405.30	±	169.40	341.10	±	145.70
**EPO**	354.00	±	206.00	146.00	±	84.4	229.00	±	251.00	256.00	±	242.00
**Osteonectin**	153.00	±	54.30	175.00	±	44.00	171.00	±	60.00	199.00	±	78.90
**Myostatin**	5.999	±	14.156	8.824	±	12.982	11.580	±	18.253	11.599	±	21.667
**Oncostatin**	2.4	±	3.8	1.29	±	1.1	3.0	±	6.0	3.8	±	7.7
**FSTL1**	491	±	297	198	±	117	464	±	299	624	±	579
**Irisin**	746	±	601	367	±	295	353	±	445	341 *	±	343
**Osteocrin**	7.93	±	3.44	5.48	±	3.89	16.40	±	17.40	17.00	±	13.80

Results are expressed as mean ± standard deviation (SD). EPO, Erythropoietin; FSTL1, Follistatin-related protein 1. * *p* < 0.05 rest vs. post-exercise.

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
