# Peer review of "Seawater Hydration Modulates IL-6 and Apelin Production during Triathlon Events: A Crossover Randomized Study"

_ijerph, 2022, doi:10.3390/ijerph19159581_

Round 1
Reviewer 1 Report
The paper by González Acevedo et al addresses the potential benefits of administering seawater for achieving a rapid and efficient hydration after demanding physical exercise (triathlon). This is an interesting and important topic because it is essential to independently support or contradict the commercial claims of manufacturing laboratories. Although important and innovative, the study needs significant improvements before publication.
1.- The manuscript requires a comprehensive and extensive editing. The English needs significant improvements because there are numerous grammar errors and many sentences are difficult to understand (for instance, the title is in blatant contradiction with the results). The abstract is confusing and data are presented in an inconsistent way (for instance, figures 1 and 2 represent “SG” for the experimental group, whereas there are no indications for that in the figure legend, being referred instead as “EG”. The manuscript is unnecessarily long in some cases, and I believe that the overall clarity would benefit from a more concise version. Therefore, a new and more carefully written and edited version is a must.
2.- The rationale for choosing the studied biomarkers is not clear and this should be better explained. As it appears now, it seems that they studied the indicated biomarkers just because they are included in the kit used. In addition, the term “cytokine” is incorrectly used to define all molecules studied, whereas the truth is that only a minority of them are typical cytokines (i.e. paragraph 3.2). Groups of molecules should be clearly identified as myokines, hormones, cytokines and so on, and referred to accordingly. The data should be presented and interpreted according to the functions of each biomarker’s group, and not under a generic and confusing term which brings misleading conclusions because they may imply a participation of the immune system in the regulation of molecules that are not related at all with immunity.
3.- The main problem of the manuscript is that, at least in the way they are represented now, data analyses and statistical comparisons are between the basal group and the seawater group in most cases, and between the basal and the control or placebo groups in some other instances. These analyses are of no biological importance, as the meaningful biological conclusion is to determine whether seawater has (or has not) an effect over the placebo group. If there are no differences between the placebo and seawater groups, then no biological effects can be attributed to seawater. In this regard, and because of its importance, precise details should be given about the exact composition of the placebo drink received by this group of individuals (“isocaloric placebo” is confusing and not sufficient, as it could contain electrolytes and salts). Furthermore, it is necessary to indicate why such a group was needed over the so called “control” group receiving tap water (the term “control” is also confusing, as the real control is the placebo group).
4.- The authors should acknowledge that one of the most (if not the most) significant weaknesses of the study is the small number of athletes studied. Fifteen individuals per group (ten in the case of the “control” group) is certainly a small number to draw ample conclusions, and such fact should be dully recognized.
Author Response
Point-by-point answer to reviewers’ comments
Manuscript ID ijerph-1797661
Reviewer #1
We thank the reviewer for his/her comments and thorough revision of the manuscript. In an attempt to address the reviewer’s concerns and make the manuscript clearer, we have changed the highlights of study. Changes to the original manuscript are highlighted in yellow. Please be aware that page numbers in this new version do not match those in the previous one.
The paper by González Acevedo et al addresses the potential benefits of administering seawater for achieving a rapid and efficient hydration after demanding physical exercise (triathlon). This is an interesting and important topic because it is essential to independently support or contradict the commercial claims of manufacturing laboratories. Although important and innovative, the study needs significant improvements before publication.
Reply: Thank you for your comment.
1.- The manuscript requires a comprehensive and extensive editing. The English needs significant improvements because there are numerous grammar errors and many sentences are difficult to understand (for instance, the title is in blatant contradiction with the results). The abstract is confusing and data are presented in an inconsistent way (for instance, figures 1 and 2 represent “SG” for the experimental group, whereas there are no indications for that in the figure legend, being referred instead as “EG”. The manuscript is unnecessarily long in some cases, and I believe that the overall clarity would benefit from a more concise version. Therefore, a new and more carefully written and edited version is a must.
Reply: Thank you for your comment. According to the reviewer's suggestions, we have sent our manuscript for correction in English, and we have also corrected the mistakes in figures 1 and 2.
2.- The rationale for choosing the studied biomarkers is not clear and this should be better explained. As it appears now, it seems that they studied the indicated biomarkers just because they are included in the kit used. In addition, the term “cytokine” is incorrectly used to define all molecules studied, whereas the truth is that only a minority of them are typical cytokines (i.e. paragraph 3.2). Groups of molecules should be clearly identified as myokines, hormones, cytokines and so on, and referred to accordingly. The data should be presented and interpreted according to the functions of each biomarker’s group, and not under a generic and confusing term which brings misleading conclusions because they may imply a participation of the immune system in the regulation of molecules that are not related at all with immunity.
Reply: Thank you for your comment. We fully agree with the reviewer, and are very grateful that he noticed this issue. This mistake has been fixed.
3.- The main problem of the manuscript is that, at least in the way they are represented now, data analyses and statistical comparisons are between the basal group and the seawater group in most cases, and between the basal and the control or placebo groups in some other instances. These analyses are of no biological importance, as the meaningful biological conclusion is to determine whether seawater has (or has not) an effect over the placebo group. If there are no differences between the placebo and seawater groups, then no biological effects can be attributed to seawater. In this regard, and because of its importance, precise details should be given about the exact composition of the placebo drink received by this group of individuals (“isocaloric placebo” is confusing and not sufficient, as it could contain electrolytes and salts). Furthermore, it is necessary to indicate why such a group was needed over the so called “control” group receiving tap water (the term “control” is also confusing, as the real control is the placebo group).
Reply: Thank you for your comment. Once again, we appreciate the interesting comment received by the reviewer. Indeed, this comment should have been included in the main text of the manuscript. For this reason, we have included a new statistical process and added it to the results. I hope that the new results added will be to the reviewer's liking.
4.- The authors should acknowledge that one of the most (if not the most) significant weaknesses of the study is the small number of athletes studied. Fifteen individuals per group (ten in the case of the “control” group) is certainly a small number to draw ample conclusions, and such fact should be dully recognized.
Reply: Thank you for your comment. In line with the reviewer's suggestions, we have added this issue to the limitations section.

Reviewer 2 Report
In line 26, it says "While seawater was not ergogenic it significantly increased" What did it increase? Is it a mistake?
In this study, author justify that athletes have limited access to fluid intake during the competition. However, the research is done with Totum seawater which is a "bottled beverage" and athletes would have limited access to it during competition.
Line 47. Authors should mention also the cons of consuming seawater as it is not filtered and is too salt-satured.
The introduction si well written and well conducted. However, the abstract started introducting the problem about the difficulty to get fluids during a triathlon, now the authors mention that the aim is to analyze the recovery after the triathlon with the use of SW (lines 84-85). Authors must clarify which is the problem to solve. In the introduction you mention some benficial aspects of SW, I think that you must also mention the harmful effects of non-filtered SW intake. You also mention the use of desalinated water from 662m below sea level, is it interesting that use during triathlon? is that water available to intake?
Line 93. Did you mean 3 triathlons in the last year? 3 triathlons in the last month almost means one per week.
Pines 95-96. The sentence is not finished ". In the present experimental 95 crossover study, fifteen federated triathletes from different clubs in Andalusia (Spain)."
115. Can you detail the standardize breakfast?
Did you finally have 25 subjects in total?
In the tables, you are using commas (,) as a decimal separator. In english you must use dots (.) as decimal separator
Author Response
Point-by-point answer to reviewers’ comments
Manuscript ID ijerph-1797661
Reviewer #2
We thank the reviewer for his/her comments and thorough revision of the manuscript. In an attempt to address the reviewer’s concerns and make the manuscript clearer, we have changed the highlights of study. Changes to the original manuscript are highlighted in yellow. Please be aware that page numbers in this new version do not match those in the previous one.
In line 26, it says "While seawater was not ergogenic it significantly increased" What did it increase? Is it a mistake?
Reply: Thank you for your comment. In this sentence what we mean is that there was no improvement in sporting performance but there was a greater release of values of IL-6, IL-15, apeline, BDNF and FGF-21 cytokines post-exercise.
In this study, author justify that athlete have limited access to fluid intake during the competition. However, the research is done with Totum seawater which is a "bottled beverage" and athletes would have limited access to it during competition.
Reply: Thank you for your comment. Indeed, triathletes have limited access to drinks for rehydration. For this reason, our beverage (totum), like the placebo and tap water, was consumed before the sport, except for the bike section which was consumed during exercise, but not during the swim and run.
Line 47. Authors should mention also the cons of consuming seawater as it is not filtered and is too salt-satured.
Reply: Thank you for your comment. However, I should point out to the reviewer that the seawater was indeed filtered; in fact, it is done in a complex way, explained in the methods section below:
The microfiltered and sterilised seawater “Totum Sport” was provided by Laboratoires Quinton Co. (Alicante, Spain). This water has been extracted from specific areas of the Atlantic Ocean, in the Bay of Biscay, at a distance of more than 10 miles from the coast and 20 meters deep, at the centre of proliferations or plankton blooms.
This sentence was added to the main document.
The introduction si well written and well conducted. However, the abstract started introducting the problem about the difficulty to get fluids during a triathlon, now the authors mention that the aim is to analyze the recovery after the triathlon with the use of SW (lines 84-85). Authors must clarify which is the problem to solve. In the introduction you mention some benficial aspects of SW, I think that you must also mention the harmful effects of non-filtered SW intake. You also mention the use of desalinated water from 662m below sea level, is it interesting that use during triathlon? is that water available to intake?
Reply: Thank you for your comment. The authors of this study do not understand why we should include in the introduction the adverse effects of consuming unfiltered seawater, given that our seawater (totum) was filtered, as indicated in the text of the main document. I think we have confused the reviewer and therefore we would like to apologize for the misunderstanding. Besides, as the reviewer rightly points out, it is true that we began the introduction by talking about the difficulty of maintaining hydration during the swimming and running phases of a triathlon event. However, our study focuses primarily on the effects of seawater on recovery after a triathlon event. For this reason, we are going to change the first paragraphs of the abstract. Therefore, it would be a good idea to check whether the consumption of this seawater could be of ergogenic help to obtain a better post-exertion recovery.
Regarding the interest of using desalinated water at 662 meters below sea level in sporting events such as a triathlon, as well presented by Hou et al., the authors argue that the consumption of this water can substantially accelerate the recovery of aerobic power and lower body muscle power after a prolonged exercise session, such as a triathlon.
Line 93. Did you mean 3 triathlons in the last year? 3 triathlons in the last month almost means one per week.
Reply: Thank you for your comment. Basically what we mean is that during the triathletes' competitive calendar, they should have done at least 3 triathlons per year. However, in order to be able to participate in the study, no triathlon should have taken place in the month prior to the experimental design, as this could alter the results of the study.
Pines 95-96. The sentence is not finished ". In the present experimental 95 crossover study, fifteen federated triathletes from different clubs in Andalusia (Spain)."
Reply: Thank you for your comment. This issue was corrected in the main document.
- Can you detail the standardize breakfast?
Reply: Thank you for your comment. According to suggestion of review, we have added the follow sentence:
For the morning session, participants came to the swimming pool after a standardized breakfast (57% carbohydrate, 18% protein and 25% fat; thus, the breakfast met the general recommendations of international institutions) ...
Did you finally have 25 subjects in total?
Reply: Thank you for your comment. The organization of the participants was as follows, the experimental group was composed of 15 triathletes. These same athletes carried out the placebo group. And of these 15 triathletes 10 were the control group.
In the tables, you are using commas (,) as a decimal separator. In English you must use dots (.) as decimal separator
Reply: Thank you for your comment. This issue was corrected in the main document.

Round 2
Reviewer 1 Report
The revised version of the manuscript has been greatly improved. However, there are some issues that still require correction prior publication.
1. The most relevant aspect that needs extensive and careful re-writing is the incorrect conclusions of the molecules modulated specifically by sea water. If the authors make such claim, then they should refer only to IL6 and Apelin, because those are the only molecules with statistical differences over the placebo group. Whereas other molecules are also modulated in the placebo group, that effect could be related to the consumption of salt (saline over tap water). Because of that, the authors should refrain from sea water claims beyond IL6 and Apelin. In this regard, the discussion should be revised accordingly and the title as well.
2. The term "isocaloric" used in this contex for the placebo is irrelevant as the key factor is the salt balance administered and not calories. The authors should refer to "saline solution (9 gr/l NaCl) was used as placebo".
3.- In line 314, the authors state that LIF is induced only by seawater. This is incorrect, as the only statistical difference is observed between the basal and tap water groups. LIF modulation should therefore be revised.
4.- The conclusion should also be revised and re-written to reflect the true nature of results, because it is not correct to say that "several myokines" are modulated by sea water. The reality is, again, that only two molecules are regulated specifically by sea water. Therefore, more restrictive conclusions should be stated.
Author Response
Point-by-point answer to reviewers’ comments
Manuscript ID ijerph-1797661
Reviewer #1
We thank the reviewer for his/her comments and thorough revision of the manuscript. In an attempt to address the reviewer’s concerns and make the manuscript clearer, we have changed the highlights of study. Changes to the original manuscript are highlighted in yellow. Please be aware that page numbers in this new version do not match those in the previous one.
The revised version of the manuscript has been greatly improved. However, there are some issues that still require correction prior publication.
- The most relevant aspect that needs extensive and careful re-writing is the incorrect conclusions of the molecules modulated specifically by sea water. If the authors make such claim, then they should refer only to IL6 and Apelin, because those are the only molecules with statistical differences over the placebo group. Whereas other molecules are also modulated in the placebo group, that effect could be related to the consumption of salt (saline over tap water). Because of that, the authors should refrain from sea water claims beyond IL6 and Apelin. In this regard, the discussion should be revised accordingly and the title as well.
Reply: Thank you for your comment. In line with the reviewer's suggestions, we have carried out all the recommended comments, both in the title and in the discussion.
2. The term "isocaloric" used in this contex for the placebo is irrelevant as the key factor is the salt balance administered and not calories. The authors should refer to "saline solution (9 gr/l NaCl) was used as placebo".
Reply: Thank you for your comment. This issue has been corrected.
3.- In line 314, the authors state that LIF is induced only by seawater. This is incorrect, as the only statistical difference is observed between the basal and tap water groups. LIF modulation should therefore be revised.
Reply: Thank you for your comment. This issue was corrected.
4.- The conclusion should also be revised and re-written to reflect the true nature of results, because it is not correct to say that "several myokines" are modulated by sea water. The reality is, again, that only two molecules are regulated specifically by sea water. Therefore, more restrictive conclusions should be stated.
Reply: Thank you for your comment. In accordance with the reviewer's suggestions, we have changed the conclusion.

Reviewer 2 Report
The authors addressed al my concerns and the article is suitable for publication in the IJERPH
Author Response
Thank you very much for the comments received, they have been of huge help.